# Fabrication of Black Body Grids by Thick Film Printing for Quantitative Neutron Imaging

**DOI:** 10.3390/jimaging8060164

**Published:** 2022-06-08

**Authors:** Martin Wissink, Kirk Goldenberger, Luke Ferguson, Yuxuan Zhang, Hassina Bilheux, Jacob LaManna, David Jacobson, Michael Kass, Charles Finney, Jonathan Willocks

**Affiliations:** 1Energy Science and Technology Directorate, Oak Ridge National Laboratory, Oak Ridge, TN 37830, USA; kassmd@ornl.gov (M.K.); finneyc@ornl.gov (C.F.); willocksj@ornl.gov (J.W.); 2Thick Film Technologies, Everett, WA 98204, USA; kirk@thickfilmtech.com; 3C12 Advanced Technologies, LLC, Everett, WA 98204, USA; info@c12materials.com; 4Neutron Sciences Directorate, Oak Ridge National Laboratory, Oak Ridge, TN 37830, USA; zhangy6@ornl.gov (Y.Z.); bilheuxhn@ornl.gov (H.B.); 5National Institute of Standards and Technology, Gaithersburg, MD 20899, USA; jacob.lamanna@nist.gov (J.L.); david.jacobson@nist.gov (D.J.)

**Keywords:** neutron imaging, black body grids, quantitative imaging, scattering correction

## Abstract

Neutron imaging offers deep penetration through many high-Z materials while also having high sensitivity to certain low-Z isotopes such as ^1^H, ^6^Li, and ^10^B. This unique combination of properties has made neutron imaging an attractive tool for a wide range of material science and engineering applications. However, measurements made by neutron imaging or tomography are generally qualitative in nature due to the inability of detectors to discriminate between neutrons which have been transmitted through the sample and neutrons which are scattered by the sample or within the detector. Recent works have demonstrated that deploying a grid of small black bodies (BBs) in front of the sample can allow for the scattered neutrons to be measured at the BB locations and subsequently subtracted from the total measured intensity to yield a quantitative transmission measurement. While this method can be very effective, factors such as the scale and composition of the sample, the beam divergence, and the resolution and construction of the detector may require optimization of the grid design to remove all measurement biases within a given experimental setup. Therefore, it is desirable to have a method by which BB grids may be rapidly and inexpensively produced such that they can easily be tailored to specific applications. In this work, we present a method for fabricating BB patterns by thick film printing of Gd_2_O_3_ and evaluate the performance with variation in feature size and number of print layers with cold and thermal neutrons.

## 1. Introduction

### 1.1. Quantitative Neutron Imaging with Black Body Grids

Neutron imaging and tomography are powerful non-destructive techniques which are very complementary to the more conventional X-ray imaging due to the unique properties of neutrons and their interactions with matter. Neutrons interact primarily with the nucleus (rather than the much larger electron cloud, as with X-rays) and therefore have high penetration through even high-Z materials. The nuclear interaction is isotope-specific, which fortuitously provides high contrast to certain low-Z materials including ^1^H, ^3^He, ^6^Li, and ^10^B and allows for isotopic labelling to generate contrast (such as between ^1^H_2_O and ^2^H_2_O). The cold and thermal neutrons typically employed for imaging also have low energy (0.1–100 meV) compared to X-rays (100–10 MeV), which makes them non-destructive/non-invasive even in low-Z materials. This unique combination of properties has made neutron imaging and tomography valuable tools across a range of disciplines including manufacturing, materials science, biology, archaeology, energy, and transportation [1].

Neutron imaging is a form of transmission radiography which measures the change in radiation intensity as it passes through an object to infer the properties of the object. The transmitted intensity of a beam with wavelength *λ* is given by the Beer–Lambert law:(1)$$\frac{I\left(\lambda \right)}{I_{0}\left(\lambda \right)}=e^{-\Sigma \left(\lambda \right)d}$$
where I is the intensity of the transmitted beam, $I_{0}$ is the intensity of the incident beam, Σ is the macroscopic linear attenuation coefficient, and d is the path length. The macroscopic coefficient Σ is determined by the microscopic cross sections $\sigma_{tot}$ of the various isotopes i which make up the material:(2)$$\Sigma \left(\lambda \right)=\overset{n}{\underset{i}{\sum }}\sigma_{tot}{\left(\lambda \right)}_{i}N_{i}$$
(3)$$N_{i}=\frac{\rho_{i}N_{A}}{M_{i}}$$
where N is the atomic density, ρ is the mass density, $N_{A}$ is Avogadro’s number, and M is the atomic mass number.

The information in a neutron image (radiograph) is then ideally a two-dimensional projection of the path-integrated attenuation coefficient of the sample, while a tomographic reconstruction combines projections from many angles to create a three-dimensional representation of the attenuation coefficient in discrete volumes of the sample. In practice, however, both radiography and tomography are generally qualitative due to a variety of biases which may contribute to the recorded images. In many cases, such as when identifying the location of interfaces between phases with high contrast, the distinction between qualitative and quantitative attenuation coefficients may be irrelevant. However, if one desires to determine the concentration or the absolute amount of constituents in the sample, quantitative attenuation coefficients are essential.

Some of the of biases that can affect the measured images are illustrated in Figure 1. *Background scattering* includes neutrons that are scattered into the detector from the surroundings or the sample environment and neutrons or converted particles which are scattered inside the detector. In the case of a scintillator-camera detector, this may include neutrons scattered within the scintillator media or the mirror and photons reflected by the mirror. Bias may also be introduced by *sample scattering*, as the total cross section is the sum of the coherent scattering, incoherent scattering, and absorption cross sections:(4)$$\sigma_{tot}=\sigma_{coh}+\sigma_{incoh}+\sigma_{abs}$$

One of the assumptions in the Beer–Lambert law is that the total cross section describes the rate at which neutrons exit the transmitted beam. However, if the neutrons are scattered in the forward direction, they may still be measured somewhere on the detector, leading to a reduction in the apparent attenuation coefficient at that location. This is particularly relevant with hydrogenous materials due to the high incoherent scattering cross section of ^1^H. Sample scattering can be addressed by moving the sample away from the detector [2], as the intensity of the incoherent scattering is inversely proportional to distance squared, but this will also reduce the spatial resolution of the image and does not address the background scattering.

Monte Carlo methods can be used to estimate scattering contributions, but these require *a priori* knowledge of both the sample and the imaging setup and are therefore difficult to apply in general cases [3,4,5,6,7]. An experimental method of correcting for background and sample scattering has recently been proposed and demonstrated by researchers at the Paul Scherrer Institute [8,9,10]. This involves deploying a grid of neutron absorbers, or *black bodies* (BBs), in front of the sample position to create a set of reference images. The signal measured behind the BBs is assumed to be due to the contributions other than transmission (i.e., scattering and other biases). By interpolating between the BB locations, a 2D “scattering image” is obtained, which can then be subtracted from the images without the BBs to obtain an estimate of the “true” transmission image, as illustrated in Figure 1C.

While the BB grid approach has been demonstrated to be very effective at removing scattering-induced artifacts in computed tomography (CT) reconstructions [8,9,10], there are some caveats. Alignment of the grid relative to certain features in the sample is critical and misplacing the grid can induce larger biases than existed to start with. Samples which have heterogeneous, highly-scattering features over a range of length scales will not be well served by a homogeneous grid. The scale of the grid required will also depend on the field of view and resolution of the detector, and imaging setups which feature adjustable magnification would therefore, ideally, have access to a range of grid sizes. It is, therefore, desirable to have a means of optimizing the design of the grid to specific combinations of sample and imaging setup, and this is only practical if prototype grids can be produced rapidly and at low cost.

### 1.2. Candidate Materials for BB Grids

The palette of materials which strongly absorb neutrons is relatively limited. Commonly employed isotopes (with cross sections at 2 Å) are ^157^Gd (2.75 × 10^−19^ cm^2^) and ^10^B (4.3 × 10^−21^ cm^2^). Ideally, the BB grid should be as thin as possible to minimize artifacts at the edges of the BBs from imperfect fabrication or alignment and to minimize physical interference with the sample environment. Therefore, it is desirable to have the highest total attenuation coefficient, which depends on the absorption cross section, the relative abundance of the absorbing isotope in the material, and the total density of the material. Isotopically enriched materials are generally quite expensive, so natural materials would be preferable for rapid prototyping or one-off applications. Here, natural Gd has a distinct advantage due to high abundance of ^157^Gd (15.6%) and ^155^Gd (14.8%), which are both strong neutron absorbers. Attenuation coefficients (at 2 Å) for natural Gd (1600 cm^−1^) and Gd_2_O_3_ (1390 cm^−1^) compare favorably with B_4_C (94 cm^−1^) and even ^10^B_4_C (500 cm^−1^). Gd sputtering has been used to create resolution masks [11] and coded apertures [12] for neutron imaging, but sputtering is a slow process and building up a layer thick enough to completely attenuate thermal neutrons is both technically challenging and costly. Gd is also flammable (or even pyrophoric) in the form of fine powder or thin foil and may react with water in the atmosphere to evolve flammable hydrogen gas, so alternative fabrication techniques such as stamping or laser cutting would require the expense and complication of an inert environment. In comparison, Gd_2_O_3_ powder is an eye and respiratory irritant but is otherwise relatively benign and is, therefore, preferable to Gd from a safety and material handling perspective. Gd_2_O_3_ is also widely available in a range of nano- and micro-scale powders and is considerably less expensive than comparably sized Gd powders.

Sufficient thickness of Gd_2_O_3_ is key to achieving high enough attenuation of the transmitted neutrons to make a reliable scattering measurement. Energy-resolved total attenuation coefficients (Σ) were calculated for Gd_2_O_3_ using NEUIT [13] with the ENDF/B-VIII.0 database [14] and are shown in Figure 2 along with the spectra at the High Flux Isotope Reactor (HFIR) CG-1D and NIST Center for Neutron Research (NCNR) BT2 neutron imaging instruments. CG-1D has a cold neutron spectrum and was measured in 2010 with a peak at 2.75 Å [15]. Compared against a Maxwell–Boltzmann distribution with the same peak wavelength, we observe that the measured CG-1D spectrum has considerably less neutrons at longer wavelengths (<10 meV). BT2 has a thermal neutron spectrum with a Maxwell–Boltzmann distribution at 1.8 Å. The required thickness to achieve attenuation of 99% and 99.9% was calculated for each of the spectra and is reported in Table 1. For monochromatic neutrons at 1.8 Å and 2.75 Å, the required thickness is similar due to the relatively flat slope of Σ for Gd_2_O_3_ between those wavelengths. However, the value of Σ drops quickly above energies of ~50 meV (λ < 1.28 Å), and for this reason we see that the thermal spectrum at BT2 requires ~5× the thickness to reach 99.9% attenuation relative to CG-1D. However, detection efficiency tends to decrease as neutron energy increases, so the *effective* spectrum may not be quite so difficult to block. In any case, to ensure at least 99% attenuation at either instrument, Gd_2_O_3_ thickness >80 μm will be required.

### 1.3. Thick Film Printing

Thick film printing is a screen printing process used to produce hybrid microelectronic circuits and can typically produce layers of thickness ranging from 0.1 μm to 100 μm [18]. Thick film is an additive technology, in that layers are built up sequentially only in the desired areas rather than depositing material uniformly and then selectively removing it by etching, as is performed with standard printed circuit boards. An overview of the process is illustrated in Figure 3. A paste is first prepared for the desired film material (conductor, resistor, or dielectric). The paste is typically composed of fine particles of the active material, a glass frit which bonds the film to the substrate, and an organic vehicle which gives the paste the desired rheological properties for screen printing. A fine screen mesh is prepared with a UV-sensitive emulsion, and the desired pattern is produced on the screen with a photographic process. The screen is suspended in a taut frame above the substrate, and a flexible squeegee is pulled across the screen, forcing the paste through the open areas and producing a pattern on the substrate. The resulting wet paste pattern is then dried at temperatures up to 150 °C and fired at temperatures up to 1000 °C to remove the organic carrier, develop the desired electrical properties, and bond the film to the substrate.

## 2. Materials and Methods

### 2.1. Fabrication of BB Grids

An organic binder system was developed to create a printable paste using natural Gd_2_O_3_ powder. The Gd_2_O_3_ was sourced from Sigma-Aldrich (Certain trade names and company products are mentioned in the text or identified in an illustration in order to adequately specify the experimental procedure and equipment used. In no case does such identification imply recommendation or endorsement by the National Institute of Standards and Technology, nor does it imply that the products are necessarily the best available for the purpose.) with a 99.9% assay (trace metals basis) and an average particle size of 1 µm to 4 µm with >90% of particles <10 µm. The vehicle for the paste consists mainly of butyl carbitol acetate solvent with a soluble ethylcellulose binder. Other paste components include small additions of a commonly available ethoxylated nonylphenol wetting agent and a dispersant (2-furoic acid) that help make the paste suitable for screen printing. Due to the high melting point of Gd_2_O_3_ (2420 °C), the paste formulation was designed to be dried rather than fired, and the printed patterns are therefore expected to be rather delicate.

Two BB grid patterns were designed, one consisting of 250 µm diameter dots on a 2.5 mm × 2.5 mm grid (11 dots × 11 dots) and the other 500 µm diameter dots on a 5 mm × 5 mm grid (6 dots × 6 dots). For both patterns, the dots occupy nominally 0.79% of the image area. As shown in Figure 4, each pattern was printed on a 38.1 mm × 38.1 mm × 1.59 mm (1.5 in × 1.5 in × 1/16 in) GE 124 fused quartz (SiO_2_) substrate. This substrate material was chosen due to high transmission to both neutrons (see Figure 2) and visible light, such that the integrity of the pattern could be visually verified after transport and handling and to enable coarse alignment of the grid by eye. To protect the printed pattern, a 0.5 mm thick Al_2_O_3_ window frame was placed around it, and another GE 124 fused quartz plate with the same dimensions as the substrate was used as a cover plate. The window frame and cover plate were attached using an electronics grade non-conductive adhesive (Resin Labs EP1200) to create an environmentally sealed assembly.

A standard thick film screen printing approach was used to deposit the patterns. For both pattern types, a 325 mesh stainless steel screen (50.2 µm opening, 27.9 µm wire diameter, 50.2% open area) was prepared with a 12.5 µm thick PEF2 emulsion. An 80 durometer (Shore A) urethane squeegee was used. The printing speed was ~100 mm/s with an initial snap-off distance (screen height above substrate) of ~1 mm. An HMI 485 screen printer was used in “Print/Flood” mode. To build up the print thickness, successive layers were deposited using the following procedure:Print Gd_2_O_3_ paste;Dry at 125 °C to 150 °C for 15 min;Increase snap-off distance as needed to accommodate increased print thickness;Repeat.

For both patterns, BB grids were prepared with varying number of print layers, ranging from two to six. The resulting print thickness reported in Table 2 is an average of three readings of random dots on each substrate measured using a Zeiss light section microscope. For both patterns, diminishing returns in print thickness are seen after four and five print layers, and visual inspection indicated that the definition of the pattern degraded for >four print layers.

### 2.2. Neutron Imaging Configurations

The BB grids were evaluated using the thermal neutron imaging instrument at the NCNR beamline BT2 and the cold neutron imaging instrument at HFIR beamline CG-1D.

Measurements at BT2 were performed using a gadolinium oxysulfide scintillator coupled to an Andor NEO sCMOS camera (2560 px × 2160 px) with a 45° mirror. The effective pixel pitch was 6.5 µm with a field of view of ~16.6 mm × 14.0 mm [19]. A 3 mm aperture was used with an aperture-to-detector distance of 6 m, resulting in an L/D ratio of 2000. After the aperture, an evacuated flight tube prevents attenuation of the beam, which would otherwise occur if travelling through air. All images were acquired with a 60 s exposure time, resulting in a median open beam count of 112 counts/px, or ~4.4 × 10^6^ counts/(cm^2^∙s). Longer or multiple exposures would have been preferable to increase the per-pixel counts, but time limitations dictated otherwise. As shown in Figure 5A, each BB grid was held in an optical filter mount on an optical post, and the motorized sample table was used to position the BB grids in front of the detector. Since the detector field of view was smaller than the BB grids, only a portion of each grid was measured (~60% of grid area). To establish the effectiveness of the BB grids in correcting measurements from a highly scattering sample, a step wedge was constructed from 10 sheets of nominally 1 mm thick cast polymethyl methacrylate (PMMA) sourced from Goodfellow, shown in Figure 5B. The thickness of each step was measured in five locations using a micrometer with 2.5 μm accuracy. Images of the PMMA step wedge were taken with and without the thickest (six print layers) 250 µm pattern BB grid.

Measurements at CG-1D were performed using a ^6^LiF/ZnS scintillator coupled to an Andor DW936 CCD camera (2048 × 2048 px) with a 45° mirror. The effective pixel pitch was 37 µm with a field of view of ~75 mm × 75 mm [20]. A 11 mm aperture was used with an aperture-to-detector distance of 6.59 m, resulting in an L/D ratio of 599. After the aperture, a He-filled flight tube prevents attenuation of the beam, which would otherwise occur if travelling through air. The flight tube is equipped with motorized boron-nitride exit slits which control the final beam size. Since the field of view in this setup was considerably larger than the BB grids, the slits were set to two positions: in the “wide” position, the beam fully covered the detector; in the “narrow” position, the beam only covered the extent of the BB grid. All images were acquired with 60 s exposure time, with median open beam count of 311,77 counts/px, or ~3.8 × 10^7^ counts/(cm^2^∙s) (wide slits).

Only the thickest (six print layers) 250 µm and 500 µm pattern BB grids were used at CG-1D. Two different hydrogenous samples were evaluated. The first eight steps from the PMMA step wedge described above were imaged with and without the 500 µm pattern BB grid. A tomographic scan was also performed with a water column in an Al tube of 6.5 mm inner diameter and 7.0 mm outer diameter. Next, 881 projections were taken of the water column *without* a BB grid at equal angular increments from 0° to 360.8°, and 26 projections were taken *with* the 500 µm pattern BB grid at equal angular increments from 0° to 360°.

### 2.3. Image Normalization with Scattering Correction

The typical normalization approach for neutron imaging requires three images:
The “dark frame” image ($I_{DF}$), which measures the count rate of the detector system at a given exposure time (due to dark current, bias, and readout noise) with the neutron shutter closed.The measured sample image ($I_{x}^{*}$), where x denotes the sample (or $x_{i}$ for the *i*^th^ projection in a tomographic scan), which is the sum of the true sample image ($I_{x}$) and the dark frame: $I_{x}^{*}=I_{x}+I_{DF}.$
The measured “open beam” image ($I_{OB}^{*}$), which accounts for the spatial inhomegeneity of the incident beam and the detector, and which is the sum of the true open beam image ($I_{OB}$) and the dark frame: $I_{OB}^{*}=I_{OB}+I_{DF}$.

The normalized sample image is then:(5)$$\frac{I_{x}}{I_{OB}}=\frac{I_{x}^{*}-I_{DF}}{I_{OB}^{*}-I_{DF}}$$

Dose normalization should also be applied to compensate for fluctuation in the intensity of the incident beam, and an approach with BB grids is described by Carminati et al. [9]. In order to improve readability, the dose operators are not shown in the equations here, but should be assumed by the reader to be included implicity.

In the limiting case of pure transmission (no scattering or other biases), $\frac{I_{x}}{I_{OB}}=T_{x}$, where $T_{x}$ is the true normalized transmission image of the sample. In practice, however, scattering from the sample ($I_{x}^{S}$) and background ($I_{OB}^{S}$) will contribute to the measured sample and open beam images:(6)$$I_{x}^{*}=I_{x}^{T}+I_{x}^{S}+I_{DF}$$
(7)$$I_{OB}^{*}=I_{OB}^{T}+I_{OB}^{S}+I_{DF}$$

Here, superscripts *T* and *S* denote the parts of the image that are due to transmission and scattering, respectively. By using BB grids, we can estimate $I_{x}^{S}$ and $I_{OB}^{S}$ in order to recover the true transmission image:(8)$$T_{x}=\frac{I_{x}^{T}}{I_{OB}^{T}}=\frac{\left(I_{x}^{*}-I_{DF}\right)-I_{x}^{S}}{\left(I_{OB}^{*}-I_{DF}\right)-I_{OB}^{S}}$$

This is achieved by taking images of both the sample and the open beam with a BB grid placed in front of the sample position:(9)$$I_{x,BB}^{*}=I_{x,BB}^{T}+I_{x,BB}^{S}+I_{DF}$$
(10)$$I_{OB,BB}^{*}=I_{OB,BB}^{T}+I_{OB,BB}^{S}+I_{DF}$$

For the transmitted components, the image will be affected by the tranmission of the BB grid ($T_{BB}$):(11)$$I_{x,BB}^{T}=T_{BB}I_{x}^{T}$$
(12)$$I_{OB,BB}^{T}=T_{BB}I_{OB}^{T}$$

The presence of the grid will also reduce the total scattering by attenuating the incident beam, but there is no straightforward way to estimate which part of the image the scattered neutrons are removed from. We, therefore, adopt the assumption of Carminati et al. [9] that the scattering is reduced homogeneously by a scalar factor $\tau_{BB},$ which is equal to the average of $T_{BB}$ over the image area:(13)$$I_{x,BB}^{S}=\tau_{BB}I_{x}^{S}$$
(14)$$I_{OB,BB}^{S}=\tau_{BB}I_{OB}^{S}$$

Ideally, $T_{BB}$ would be 0 behind the BBs and 1 everywhere else. However, imperfect fabrication of the BBs and imperfect collimation of the incident beam may lead to some transmitted neutrons being measured behind the BBs, and the necessity of a substrate to support the BBs will cause attenuation in other regions. If we consider each BB dot to have a scalar transmission value $\tau_{dot}$, then we can estimate the scattering contribution under the BBs as
(15)$$\begin{array}{c} I_{x,BB}^{S}=\left(I_{x,BB}^{*}-I_{DF}\right)-I_{x,BB}^{T}=\left(I_{x,BB}^{*}-I_{DF}\right)-\tau_{dot}I_{x}^{T}\\ \;\;\;\;\;\;\;=\left(I_{x,BB}^{*}-I_{DF}\right)-\tau_{dot}\left[\left(I_{x}^{*}-I_{DF}\right)-I_{x}^{S}\right]\\ \;\;\;\;\;\;\;=\left(I_{x,BB}^{*}-I_{DF}\right)-\tau_{dot}\left[\left(I_{x}^{*}-I_{DF}\right)-\frac{I_{x,BB}^{S}}{\tau_{BB}}\right]. \end{array}$$

Rearranging, we obtain
(16)$$I_{x,BB}^{S}=\frac{\left(I_{x,BB}^{*}-I_{DF}\right)-\tau_{dot}\left(I_{x}^{*}-I_{DF}\right)}{1-\frac{\tau_{dot}}{\tau_{BB}}}$$

Silarly,
(17)$$I_{OB,BB}^{S}=\frac{\left(I_{OB,BB}^{*}-I_{DF}\right)-\tau_{dot}\left(I_{OB}^{*}-I_{DF}\right)}{1-\frac{\tau_{dot}}{\tau_{BB}}}$$

By substitution into Equation (8), we obtain the corrected transmission
(18)$$T_{x}=\frac{I_{x}^{T}}{I_{OB}^{T}}=\frac{\left(I_{x}^{*}-I_{DF}\right)-\frac{\left(I_{x,BB}^{*}-I_{DF}\right)-\tau_{dot}\left(I_{x}^{*}-I_{DF}\right)}{\tau_{BB}\left(1-\frac{\tau_{dot}}{\tau_{BB}}\right)}}{\left(I_{OB}^{*}-I_{DF}\right)-\frac{\left(I_{OB,BB}^{*}-I_{DF}\right)-\tau_{dot}\left(I_{OB}^{*}-I_{DF}\right)}{\tau_{BB}\left(1-\frac{\tau_{dot}}{\tau_{BB}}\right)}}.\;$$

A point to emphasize is that we cannot directly measure the scalar $\tau_{dot}$ from the images of the BB grid without having an independent measure of the scattering, because both scattered and transmitted neutrons may reach the pixels behind the BBs. Similarly, the average transmission through the substrate ($\tau_{subs}$) cannot be measured directly without first performing a scattering correction. If $\tau_{dot}$ and $\tau_{subs}$ are assumed, estimated, or measured based on fitting a calibration sample, $\tau_{BB}$ can be estimated by considering the area of the BB grid that is covered by the dots ($A_{dot}$) and the total area ($A_{total}$):(19)$$\tau_{BB}=\frac{\tau_{dot}A_{dot}+\tau_{subs}\left(A_{tot}-A_{dot}\right)}{A_{total}}$$

### 2.4. Image Processing

All image processing was performed in MATLAB^®^ [21].

#### 2.4.1. Statistical Analysis of BB Grids

The image processing pipeline for statistical analysis of the printed BB grids is demonstrated in Figure 6 for the two-layer, 250 µm pattern BB grid measurements taken at BT2. Five images were acquired for each BB grid as well as for the open beam and dark frame. As shown in Figure 6A, the raw images contain many random blob and streak artifacts. These were addressed by taking the median of each image stack to create composite images, which were then normalized according to Equation (5), with the result shown in Figure 6B. These composite normalized images are the sum of the transmitted and scattered neutrons in the sample image relative to the open beam. To aid in image binarization and segmentation, an iterative Poisson denoising algorithm was applied [22] (Figure 6C). The filtered images were then binarized using Otsu thresholding [23], and morphological cleaning was applied to remove any regions with equivalent diameter <80% of the nominal dot diameter for a given pattern and any regions in contact with the edges of the image (Figure 6D). Finally, the center of mass was calculated for each grid region, and all pixels within 2× the nominal diameter of the center were converted to radial distance and fit with an error function to compute metrics for each individual dot, as shown in Figure 6E:(20)$$y=a+\frac{\left(b-a\right)}{2}\left[1+\mathrm{erf}\left(\frac{x-r}{\sigma }\right)\right]$$

Here, r corresponds to the dot radius, σ provides an estimate of the blur at the dot edge caused by imperfection in the grid fabrication and unsharpness in the images, and a and b are the lower and upper asymptotes, respectively, which correspond to the total neutrons counted relative to the open beam at the center of the dot and in the surrounding substrate.

Each of the BB grids measured at BT2, shown in Figure 7, was processed with the pipeline described above. For the BB grids measured at CG-1D, shown in Figure 8, the images were cropped to a 750 px × 750 px region after normalization and the Poisson filter was not required for binarization, but the rest of the image processing pipeline was the same. All statistics shown for measurements at both instruments were calculated using the composite normalized (unfiltered) images.

#### 2.4.2. PMMA Step Wedge

The grid centers for the open beam images were identified using the segmentation procedure described in the previous section. For the BT2 images of the step wedge, the BB grid was in a different position than in the open beam image, and due to the variation in contrast between the BBs and the different parts of the wedge, adaptive thresholding was used in the binarization step [24]. In the CG-1D images of the step wedge, the BB grid was in the same location as in the open beam images, so the same grid centers were used for both. Using the method described in Section 1.2, the transparency of the SiO_2_ substrate (and coverplate) $\tau_{subs}$ was estimated as 92.6% for the BT2 spectrum and 91.8% for the CG-1D spectrum. Values of the dot transparency $\tau_{dot}$ were varied to investigate its impact as an adjustable parameter, and all BB images were processed according to Equations. (12)–(17) to obtain images in which the values under the BBs would correspond to $I_{x}^{S}$ and $I_{OB}^{S}$.

In the resulting images, the median value of all pixels within a two pixel radius of each grid center was assigned to that grid center, and thin plate spline interpolation (TPSI) was applied to estimate the full scattering images $I_{x}^{S}$ and $I_{OB}^{S},$ as shown in Figure 9. The low per-pixel counts in the BT2 images led to a high level of variation in the interpolated scattering images. The scattered neutron fraction of the sample image should increase monotonically with increasing wedge thickness, but several local undulations are evident for the BT2/TPSI method in Figure 9. Thin plate smoothing splines (TPSS) were also applied to the point data and produced a result more in line with expectations. The CG-1D images had much higher per-pixel counts, and the TPSI and TPSS methods had almost indistinguishable results.

#### 2.4.3. Water Column

The image processing and scattering correction steps for the water column CT scan performed at CG-1D were essentially the same as for the PMMA step wedge, except that they were repeated for each of the 26 projections taken with the BB grid. Linear interpolation was used between these to estimate the scattering at each of the 881 projections taken without the BB grid. Only the TPSI method was used for interpolation between the BBs, as TPSS consistently underestimated the scattering at the center of the water column. The scattering corrections were performed in MATLAB^®^, and CT reconstructions were performed using Muhrec [25,26].

The same processing pipeline was used in Muhrec for each of the reconstructions shown here, which included tilt correction, morphological spot cleaning, and wavelet ring cleaning [27]. Muhrec employs filtered back projection, and the same settings were used in all reconstructions. Beam hardening (BH) is a well-known issue in radiography, which stems from the decrease in attenuation coefficient with increasing radiation energy (see Figure 2), causing the average energy of the beam to increase as it passes through the sample. BH artifacts in CT reconstructions tend to appear as cupping of the Σ value throughout the material, where it is higher at the surface and lower in the interior. BH corrections were explored here via a polynomial correction module in Muhrec, which is applied to the log-normalized images (Σd). The polynomial coefficients were calculated by computing Σd as a function of path length using the energy-resolved total cross section of H_2_O [16], the measured spectrum at CG-1D, and a 2.75 Å Maxwell–Boltzmann spectrum (see Figure 2). The polychromatic values of Σd were plotted against the monochromatic value at 2.75 Å ($\Sigma =4.7{\;\mathrm{cm}}^{-1}$), and a third-order polynomial was fit to each curve. Coefficients are given in Table 3.

This approach does, in principle, introduce an error in the parts of the sample that are not water, but as the maximum value of Σd through the empty Al tube would be ~0.02, the polynomial correction has a negligible effect on the Al.

## 3. Results

### 3.1. Statistical Analysis of BB Grids

For the images of the BB grids taken at BT2, the results of the individual dot fitting are shown in Figure 10. Both the 250 µm and 500 µm patterns showed stable geometry for two to four print layers, whereas both the magnitude and spread of the blur and radius increased more dramatically at five and six print layers. At two to four print layers, the average blur for both patterns was ~50 µm, which corresponds to the size of the openings in the screen used to print the patterns. The BT2 imaging configuration has a typical resolution of $\delta_{10}=1.46\sigma ≅10\;\mathrm{to}\;20\;\mu m$, or σ≅10 μm [28]. For convolution of two Gaussian functions, the standard deviations add in quadrature:(21)$$\sigma_{f⊗g}=\sqrt{\sigma_{f}^{2}+\sigma_{g}^{2}}$$

Therefore, with our total blur $\sigma_{f⊗g}≅50\;µm$ and our imaging system blur $\sigma_{g}≅10\;\mu m$, the “physical” blur induced by the tapered edges of the dots was $\sigma_{f}≅49\;\mu m$. In other words, the contribution of the imaging configuration to the total blur of the dots was almost negligible for the images taken at BT2.

The average value of the lower asymptote for each dot, which is an estimate of the scattered background, decreased monotonically as the number of print layers increased, indicating that the dots became increasingly opaque. For the thickest 250 µm pattern, the average was ~11% of the open beam, whereas for the thickest 500 µm pattern, the average was ~7% of the open beam. Assuming that the same background exists in both measurements, this implies that the scalar transmission $\tau_{dot}$ for the dots in the 250 µm pattern was ≥4%.

For the images of the two six-layer BB grids taken at CG-1D, the results of the individual dot fitting are shown in Figure 11. Unlike the BT2 images, the larger pixel size and lower L/D ratio of the CG-1D images meant that the imaging system resolution was a significant contribution to the measured blur at the dot edges. If we consider the BT2 images as a baseline and apply Equation (21), we obtain an imaging system blur $\sigma_{g}≅70\;\mathrm{to}\;75\;\mu m$ for the CG-1D images, which is consistent with previous measurements [20,29,30]. Narrowing the exit slits on the flight tube increased the total blur by ~1.25× but also decreased the scattering background by ~4% of the open beam for the 500 µm pattern. With the wide slits, the average value for the lower asymptotes of the thickest 250 µm pattern was ~15% of the open beam, whereas for the thickest 500 µm pattern, the average was ~8% of the open beam. This implies that the scalar transmission $\tau_{dot}$ for the dots in the 250 µm pattern was ≥7%. At first glance, this seems counterintuitive, as one would expect the same BB grid to have lower transmission at CG-1D than at BT2 due to the colder spectrum. However, we can attribute this to the much larger blur in the CG-1D images.

Figure 12A,B depicts the effect of an increasingly blurry Gaussian point spread function (PSF) on the apparent transmission of a nominally opaque 250 µm diameter dot. For σ=25 µm, there is still a flat opaque region at the center of the dot, but as σ increases further, the center becomes rounded and transmission increases. This behavior is generalized in Figure 12C, where the transmission at the center of a dot of radius r is plotted as a function of σ/r. Of more practical interest is the transmission at a central *region*, as averaging multiple pixels together to improve signal/noise will be a desirable feature of any real-world BB implementation. In this case, we propose as a first-order estimate that σ~2 px and an averaging region of r≤2 px is to be used, thereby yielding r≤σ as our non-dimensional averaging region. As shown in Figure 12C, this criterion results in a higher transmission than simply using the dot center.

It is evident from Figure 12C that perfectly opaque BBs are not possible in any real imaging system. There are at least three potential ways of addressing this:

Using the determination of σ for the imaging system at the position on the BB grid, select an acceptable transparency threshold and fabricate a grid with BBs of appropriate radius. Assuming an averaging region r≤σ, a 1% transparency threshold would require σ/r<0.3, while a 0.1% threshold would require σ/r<0.24. This sets a lower bound on the BB radius, which may not be possible or desirable in all configurations. Generally, the BBs should cover as little area as possible to minimize their impact on the biases being measured. In kinetic studies where the BBs will be left in place continuously, there is also the concern of the BBs occluding interesting parts of the sample.Use the determination of the PSF for the imaging system at the position on the BB grid and perform deconvolution to recover the “true” signal at the BB centers. This approach has been demonstrated successfully with BBs [8] but has the added experimental complication of requiring detailed PSF measurements, which may not always be possible. Deconvolution may also introduce undesirable artifacts which can impact quantitative interpretation.Include the transparency of the BBs explicitly in the formulation of the scattering correction, as performed here in Equations (15)–(18) with the term $\tau_{dot}$. This approach has the advantage of allowing the imperfect opacity of the physical BB grid and the effects of the imaging system to be captured in a single term. We can bound $\tau_{dot}$ with calibration samples of known composition and dimensions, or it may also be adjusted as a free parameter to establish an uncertainty range for the corrected data.

### 3.2. PMMA Step Wedge

For each segmented region in the PMMA step wedge, the implied value of Σd was calculated for each pixel using Equation (1). The results from the BT2 images are shown in Figure 13 and include the uncorrected data as well as the BB correction with both interpolation methods. Figure 13A,B shows Σd vs. measured step thickness for $\tau_{dot}=0\%$ and $\tau_{dot}=5\%$, and the predicted values of Σd using energy-resolved cross sections for PMMA [17] and the 1.8 Å Maxwell–Boltzmann spectrum at BT2 are overlaid for reference. Figure 13C,D shows the difference between the measured and predicted values of Σd for $\tau_{dot}=0\%$ and $\tau_{dot}=5\%$. For $\tau_{dot}>5\%$; negative values are measured at one or more BB locations in the open beam image, therefore, this places an upper bound on $\tau_{dot}$.

The variance in the distributions is relatively large, even at low sample thickness, due to the low per-pixel counts, and the variance and skewness increase with thickness as the counts drop further. However, clear trends can be seen in the distribution means. The uncorrected values of Σd are always below the predicted values, and the difference becomes larger as sample thickness increases. This is consistent with expectations, as the scattering bias should become a larger fraction of the measured image as the transmission decreases. Since the data shown in Figure 13 include nearly the entire area of each step, the local undulations in scattering shown in Figure 9 are averaged out, and the scattering interpolation method has negligible impact on the results. Regardless of the value of $\tau_{dot}$ selected, the BB scattering correction achieves a much better fit to the predicted values of Σd than the uncorrected data. For $\tau_{dot}=0\%$, there is a slight overcorrection at thicknesses of 3 mm and 4 mm. For $\tau_{dot}=5\%$, the fit for 0 mm to 4 mm is excellent, but there is an undercorrection at 5 mm. This could plausibly be attributed to the much larger spread in the data at increased thickness, and longer collection times or a greater number of images would be preferable to improve the statistics in calibration images. Combined with the analysis in the previous section, we obtain an estimate of $\tau_{dot}=4\%\;\mathrm{to}\;5\%$ for the six-layer 250 µm pattern BB grid in the BT2 imaging configuration.

The PMMA step wedge results from CG-1D are shown in Figure 14, in a similar manner as the BT2 results. Figure 14A,B shows Σd vs. measured step thickness for $\tau_{dot}=0\%$ and $\tau_{dot}=3\%$, and the predicted values of Σd are overlaid for three different assumed spectra: the CG-1D spectrum as measured in 2010 and 2.75 Å and 2.9 Å Maxwell–Boltzmann spectra. The distributions have much lower variance than in the BT2 data due to higher per-pixel counts, and there is, again, no substantial difference between the two interpolation methods. Figure 14C shows the relative difference (% error) between the measured and predicted values of Σd for the matrix of $\tau_{dot}$ and spectrum combinations.

For the uncorrected data with the measured spectrum, the measured values of Σd are greater than the predicted values for steps < 5 mm thick. If the predicted values are accurate, the only way for this to occur is if the ratio of scattered to transmitted neutrons in the sample image is less than in the open beam image, or $\log_{e}\left[\frac{I_{x}^{S}}{I_{x}^{T}}/\frac{I_{OB}^{S}}{I_{OB}^{T}}\right]<0$. However, it can easily be shown that for a homogeneous sample of uniform thickness with nonzero forward scattering which covers the entire detector, $\log_{e}\left[\frac{I_{x}^{S}}{I_{x}^{T}}/\frac{I_{OB}^{S}}{I_{OB}^{T}}\right]\geq 0$. For a sample with non-uniform thickness, such as this step wedge, there is a conceivable possibility that the thicker part of the sample aligns with the detector in such a way as to cause the observed effect in thinner parts of the sample due to spatial inhomogeneity of the scattering processes within the detector. However, the data do not support this possibility, as pseudocolor images of $\log_{e}\left[\frac{I_{x}^{S}}{I_{x}^{T}}/\frac{I_{OB}^{S}}{I_{OB}^{T}}\right]$ in Figure 15 show that it is > 0 everywhere in the sample regardless of the value chosen for $\tau_{dot}$. Note that for $\tau_{dot}=3\%$, the values of this ratio are much higher, as our estimate of $I_{OB}^{S}$ tends toward 0 as $\tau_{dot}$ increases.

The other possible explanation for the measured value of Σd exceeding the predicted value is that the prediction is incorrect. The only inputs into the prediction are the energy-resolved cross sections for PMMA [17] and the measured spectrum for CG-1D [15], and the uncertainty in the cross sections is not large enough to produce this anomaly, so the only remaining conclusion is that the current spectrum has a greater cold neutron content than when last measured in 2010. If either 2.75 Å or 2.9 Å Maxwell–Boltzmann distributions are used, the measured value of Σd is less than the predicted value for all steps, and the error in the corrected data is much lower. For $\tau_{dot}=0\%$, the overall error is minimized with a 2.9 Å spectrum, whereas for $\tau_{dot}=3\%$, the error is minimized for a spectrum between 2.75 Å and 2.9 Å.

There are several factors that could plausibly justify the apparent shift in the CG-1D spectrum to colder neutrons. Refraction of neutrons within the guides is wavelength dependent, and the total flux and spectrum of neutrons is heterogenous across the cross section of the guide. In January 2014, the location of the aperture within the guide cross section was changed to maximize total flux, and this may have also shifted the spectrum. The time-of-flight spectrum measurements in 2010 were collected with a chopper and a delay-line anode detector, which used a Gd-doped microchannel plate [31] that likely has a different energy-dependent neutron detection efficiency than the ^6^LiF/ZnS scintillator used here. Other components such as an Al_2_O_3_ diffuser just after the aperture have also been added since 2010 and may affect the wavelength distribution at the detector.

### 3.3. Water Column

Horizontal and sagittal slices are shown in Figure 16 for CT reconstructions with no correction applied and with both BB and BH correction applied. The uncorrected slices exhibit significant cupping artifacts due to both scattering and beam hardening, while the corrections effectively flatten the attenuation coefficient through the sample. The corrected horizontal slice is nearly uniform, whereas the sagittal slice still exhibits some local variation.

The impact of various settings for the BB and BH corrections are explored in Figure 17, where the azimuthally averaged radial profiles are shown for horizontal slices at the same location as in Figure 16. With no BB correction and no BH correction, significant cupping is observed, and the entire distribution is below the target attenuation coefficient. Applying BH correction alone reduces the cupping but does not remove it completely. Application of BB scattering correction alone brings the average value very near to the target, but some cupping remains due to BH. The combination of BB and BH correction flattens the profile: using the measured CG-1D spectrum overshoots considerably, but the 2.75 Å Maxwell–Boltzmann spectrum matches the target value out to a radius of ~2.5 mm.

The tapering of the profile beyond 2.5 mm can be attributed to BB interpolation artifacts. Figure 18 shows the raw BB sample projection, the interpolated scattering image, and the scattering as a fraction of the sample image for the projection at 0°. At this angle, there are BBs aligned with the edges of the water column, but there are none in the center. The interpolated scattering image is smooth, and the scattered fraction of sample image generally displays the expected behavior of the fraction by increasing at thicker parts of the sample, though it is not entirely uniform. Figure 18 also shows sinograms of the raw sample projection, interpolated scattering, and scattering fraction at the horizontal slice indicated in the projections. The water column is not perfectly aligned with the rotation axis, as is observed to precess in the raw sinogram. However, the intensity distribution within the water column is consistent regardless of projection angle. The nominal edges of the water column are overlaid on the scattering sinogram, and they illustrate that while the scattering interpolation generally follows the precession of the sample, artifacts are introduced at different projection angles due to the sparse BB sampling. The BB locations are indicated by vertical lines in the scattering sinogram, while the projection angles with BBs are indicated by horizontal lines. The sampling density in the projection angle space is more than adequate to follow the movement of the sample, but the physical distance between the BBs is too large to accurately capture the shape of the scattering image. At the beginning and end of the CT scan, the BBs are aligned with the edges of the water column, but there are none in the middle, thereby underestimating scattering in the middle at those angles. Conversely, at the projection angles between roughly 120° and 300°, there is a BB within the column but none at the edge, and the scattering at the edges is thereby overestimated. This results in an inhomogeneity in the scattered fraction sinogram.

The cumulative effect in the CT reconstructions with BB correction is that Σ is overestimated in the middle of the water column and underestimated at the edges. This implies that we may need to set $\tau_{dot}>0\%$ to match the expected value of Σ if the density of the scattering sampling were to be increased. One way to increase the sampling density, as demonstrated by Boillot et al. [8], is to translate the BB grid to multiple locations. This was not pursued here due to time and resource constraints but will be an important factor in future investigations. Another method would be to create a denser BB grid, though there are tradeoffs with this approach: adding more BBs of the same radius will occlude a larger fraction of the image, which will increase the difference between the scattering that exists with and without the BBs present; adding more BBs and decreasing their radius to maintain the same covered area will reduce the effective opacity of the BBs, as demonstrated in Figure 12.

## 4. Discussion

The approach of using thick film printing to create BB grids appears to be quite promising. Test cases of BB scattering correction with a PMMA step wedge using cold and thermal neutrons and a water column CT using cold neutrons demonstrated dramatic improvement in the measured values of Σ (error reduced from >15% to <1%) when compared to the uncorrected data. The cost was also very approachable, considering that these were the first batches of prototypes. The production cost was ~$1000/grid, which included the identification and acquisition of materials, development of the paste formulation, development of the printing methodology, and the actual fabrication and delivery of the grids. This cost will likely decrease significantly if the paste is produced in larger batches and the various steps in the process move beyond the prototype stage. Areas for improvement have also been identified and are discussed below.

Several print layers were required to build up the thickness of the patterns, and this resulted in a loss of geometric fidelity. Figure 19 shows the progression of the BB shape as the number of print layers increased from two to six. What begins as a nearly cylindrical shape with sharp edges and a mounded top eventually becomes a cone. Beyond four layers, further increase in thickness is accompanied by an increase in radius and tapered edges.

Even for a low number of print layers, the desired geometry was not accurately reproduced. As shown in Figure 10 for two and three print layers, the 500 µm diameter dots were undersized by ~10 µm, and the 250 µm diameter dots were undersized by ~40 µm. This can be understood by comparing the dots to the screen used to produce them, as shown in Figure 20. It is apparent that the edges of the 250 µm dot are badly clipped by the mesh, which is too coarse to adequately produce the desired geometry.

It should be emphasized that this was a preliminary effort at thick film printing of BBs, and no attempts were made to optimize any part of the process beyond an initial best guess. For a given feature size, there are several variables that may be optimized to improve the quality of the print, including the screen size and thickness of the emulsion, the rheological properties of the paste, and the settings used during printing (such as print speed and snap-off distance). Postprocessing steps may also be applied to improve the finished result. Lapping of the surface will produce a uniform thickness, which would remove any dot-to-dot opacity variation. Lapping could also potentially be applied between print layers to reduce the tendency toward forming conical piles rather than cylinders. Laser trimming is commonly employed to adjust thick film resistors and could likely be used to clean up the edges of printed BBs [18]. A combined approach would be to intentionally oversize the features to provide a larger base, lap the surface between layers, and, finally, laser trim the edges to the desired diameter after sufficient thickness is achieved.

Beyond the fabrication of BB grids, there are other potential uses for thick film printing of Gd_2_O_3_ with neutron imaging. One would be creation of masks for samples or sample environments in which certain areas are sensitive to neutrons, such as embedded processors and memory chips, which can suffer soft errors such as bit-flipping under strong neutron flux [32]. It is common to attach neutron absorbing materials such as boron-impregnated rubber to sensitive components, but this may not be practical in all circumstances. Another potential use would be for the creation of coded apertures, which can offer substantial improvement in spatial resolution [12]. However, it is unlikely that the feature resolution achievable with thick film printing would be suitable for creating finer structures such as resolution masks or gratings.

## Figures and Tables

**Figure 1 jimaging-08-00164-f001:**
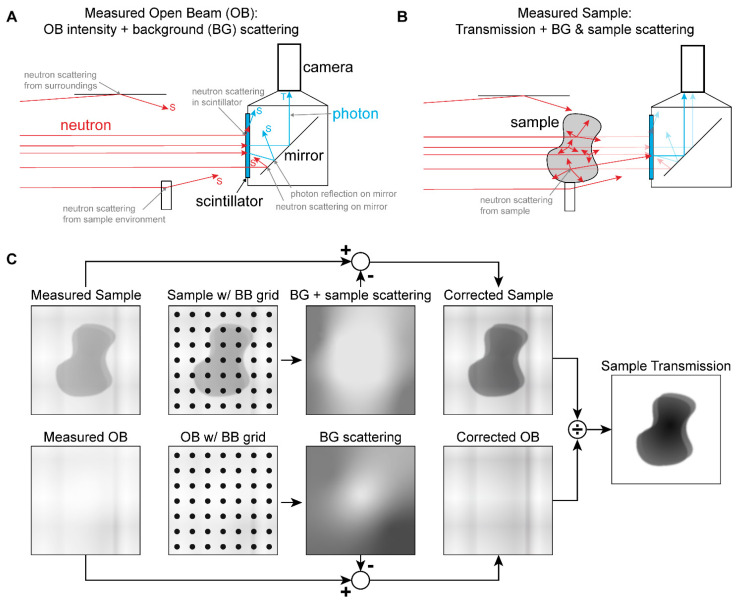
(**A**) Background scattering biases. (**B**) Sample scattering bias. (**C**) BB correction approach.

**Figure 2 jimaging-08-00164-f002:**
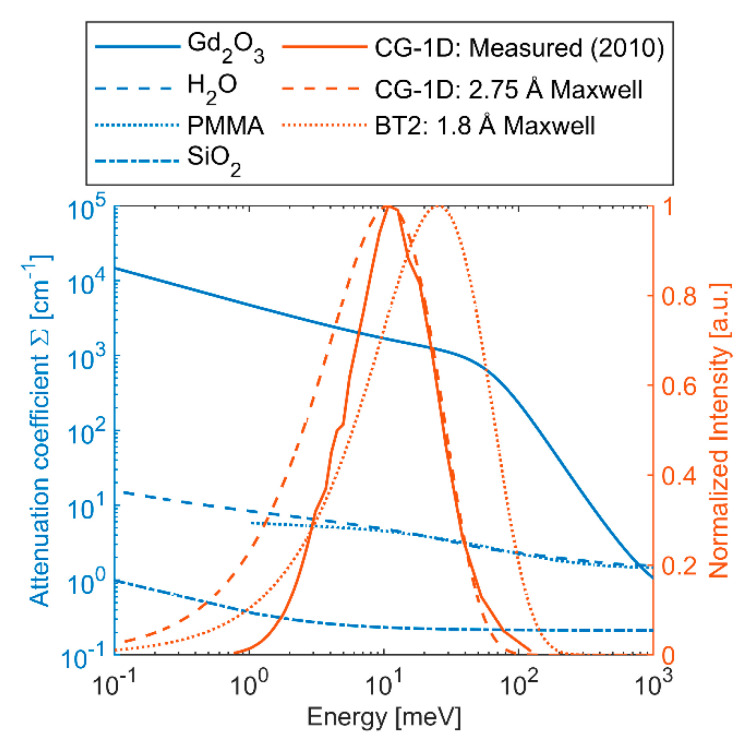
Energy-resolved attenuation coefficients for Gd_2_O_3_ [13,14], H_2_O [16], PMMA [17], SiO_2_ [13,14], and spectra for CG-1D and BT2 instruments.

**Figure 3 jimaging-08-00164-f003:**
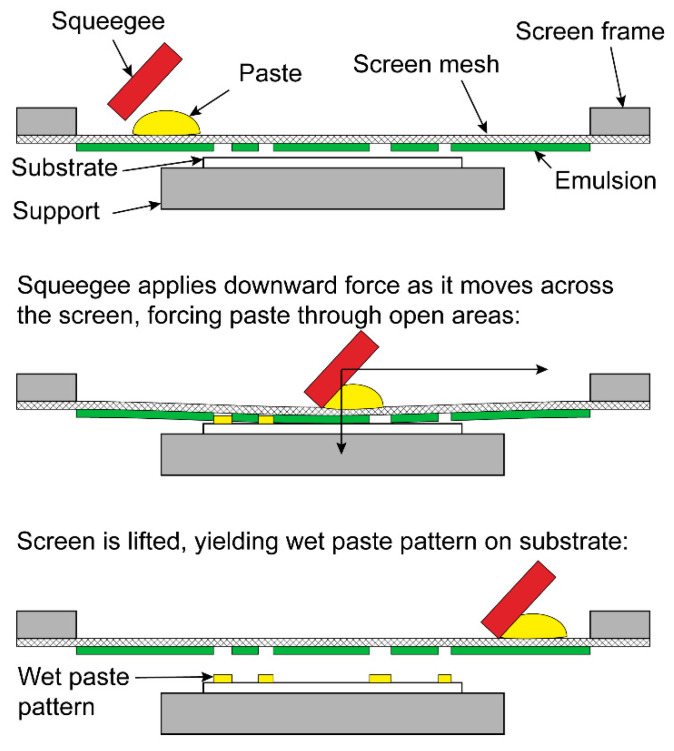
Thick film printing process.

**Figure 4 jimaging-08-00164-f004:**
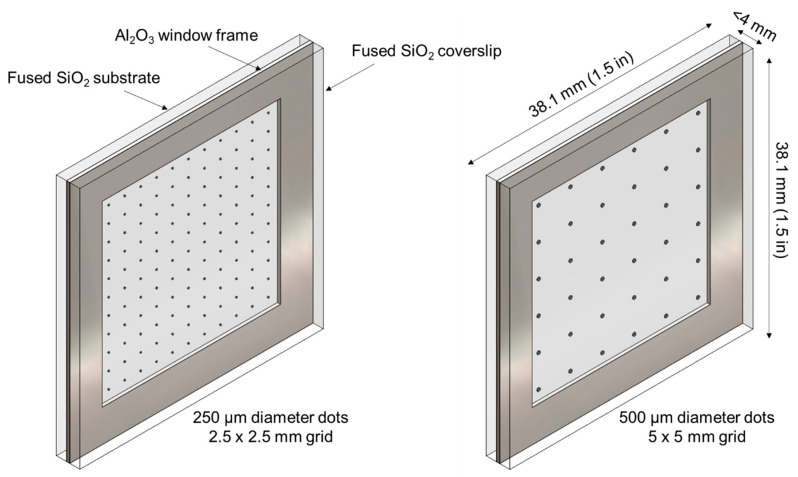
Design of environmentally sealed BB grids with thick film printing of Gd_2_O_3_ dots.

**Figure 5 jimaging-08-00164-f005:**
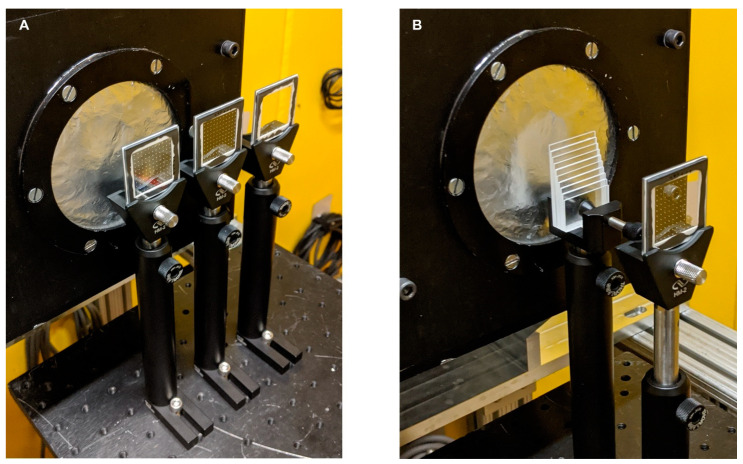
NCNR BT2 setup. (**A**) BB grids placed in optical holders in front of neutron detector. (**B**) PMMA step wedge between BB grid and detector.

**Figure 6 jimaging-08-00164-f006:**
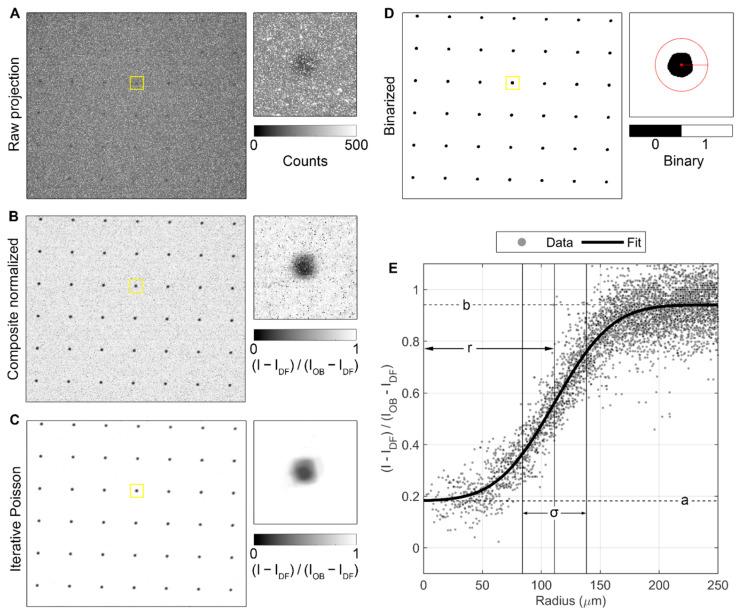
Image analysis pipeline demonstrated for the two-layer, 250 µm grid pattern measured at BT2. (**A**) Raw projections contain blob and streak artifacts. (**B**) Median of five projections each of the sample, open beam, and dark frame images are used to remove artifacts and create composite normalized images, which are the sum of the transmitted and scattered neutrons. (**C**) Iterative Poisson denoising is applied to prepare the images for binarization. (**D**) Image is binarized and regions identified and filtered by size and nearness to edge of image. (**E**) Center of mass of each grid dot from D is identified, and all pixels within 2× the nominal diameter of the center are converted to radial distance and fit with an error function to compute metrics for each individual grid dot.

**Figure 7 jimaging-08-00164-f007:**
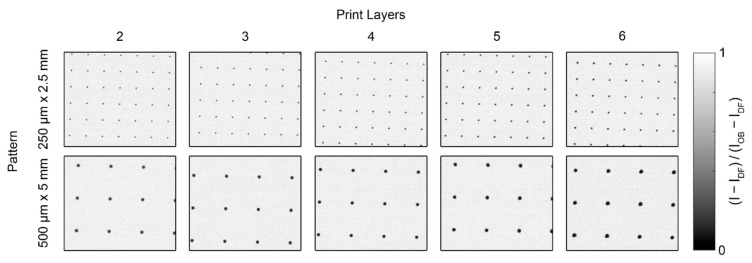
Composite normalized images of the prototype BB grids measured at BT2.

**Figure 8 jimaging-08-00164-f008:**
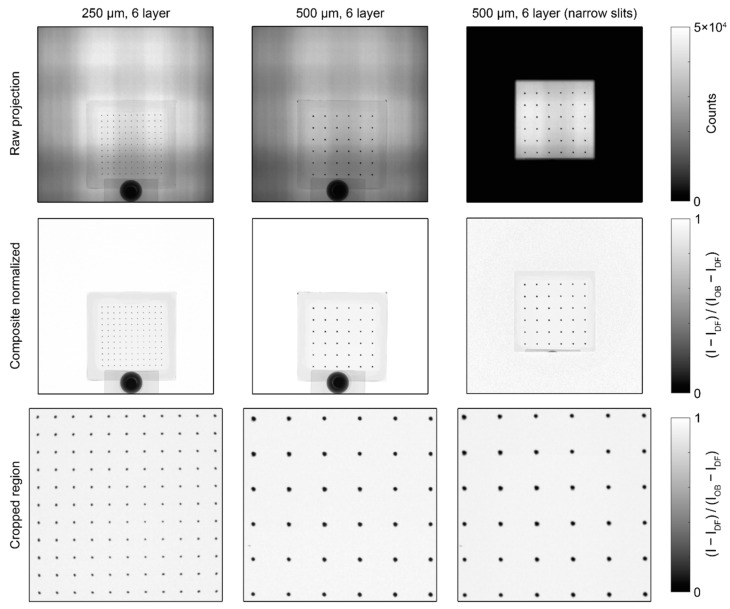
Composite normalized images of the prototype BB grids measured at CG-1D.

**Figure 9 jimaging-08-00164-f009:**
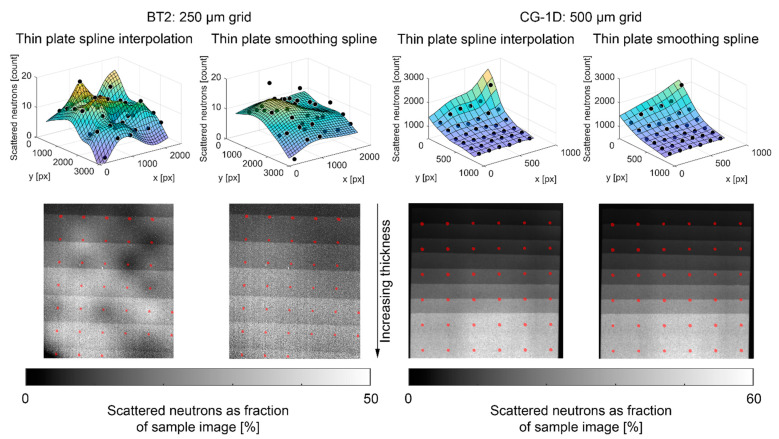
Comparison of scattering interpolation methods with PMMA step wedge. Surface plots compare interpolation methods to the measured scattering data. Black and white images show interpolated scattering as a fraction of the measured sample image. Overlaid red dots indicate locations of BBs.

**Figure 10 jimaging-08-00164-f010:**
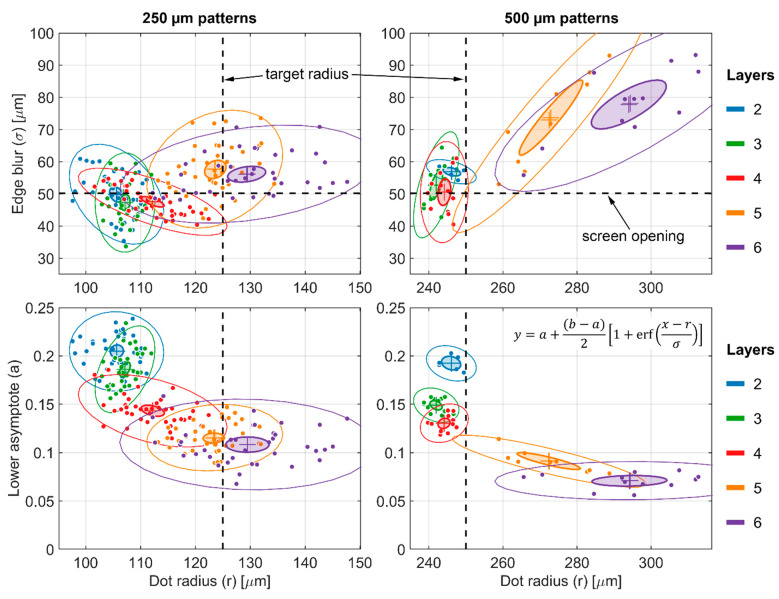
Error function fit statistics for BB dots imaged at BT2.

**Figure 11 jimaging-08-00164-f011:**
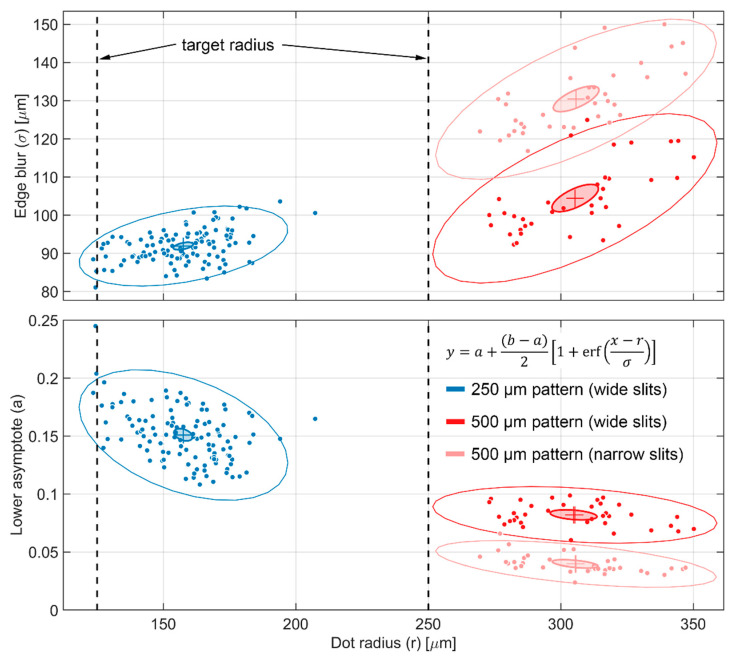
Error function fit statistics for BB dots imaged at CG-1D.

**Figure 12 jimaging-08-00164-f012:**
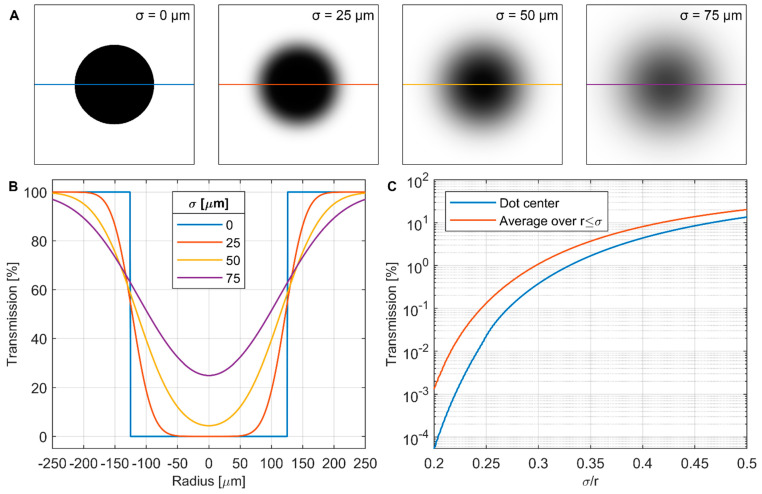
Effect of image blur on apparent transmission of a circular black body. (**A**) Model images of a perfectly opaque 250 µm diameter dot blurred with Gaussian function of increasing standard deviation σ. (**B**) Transmission profiles through each image from (**A**). (**C**) Generalized transmission at center of a dot radius r as a function of σ/r and average over a center region r≤σ.

**Figure 13 jimaging-08-00164-f013:**
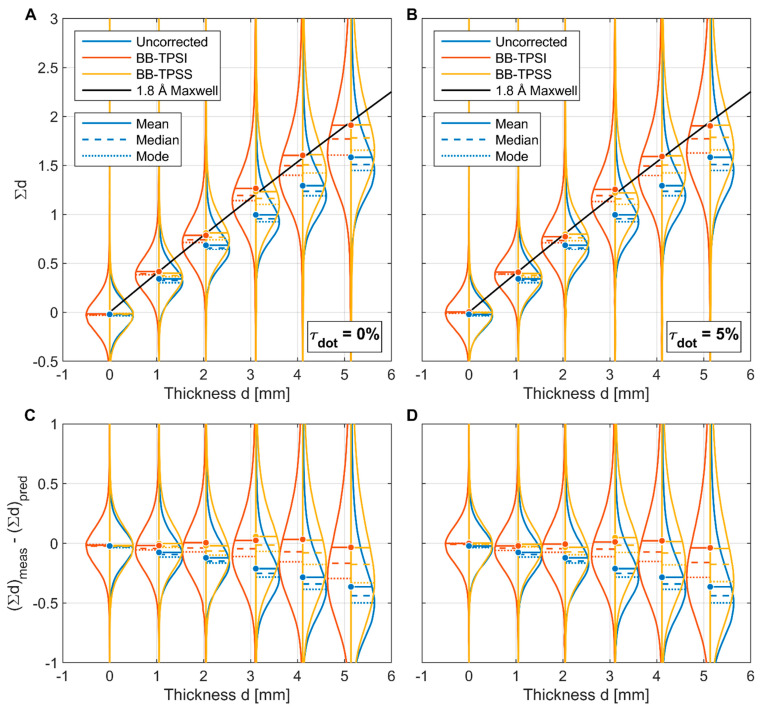
Comparison of BB scattering corrections for PMMA step wedge measured at BT2 against predicted values of Σd using known cross sections [17] and a 1.8 Å Maxwell–Boltzmann spectrum. (**A**) Values of Σd assuming 0% BB transparency. (**B**) Values of Σd assuming 5% BB transparency. (**C**). Difference between measured and predicted values of Σd assuming 0% BB transparency. (**D**). Difference between measured and predicted values of Σd assuming 5% BB transparency.

**Figure 14 jimaging-08-00164-f014:**
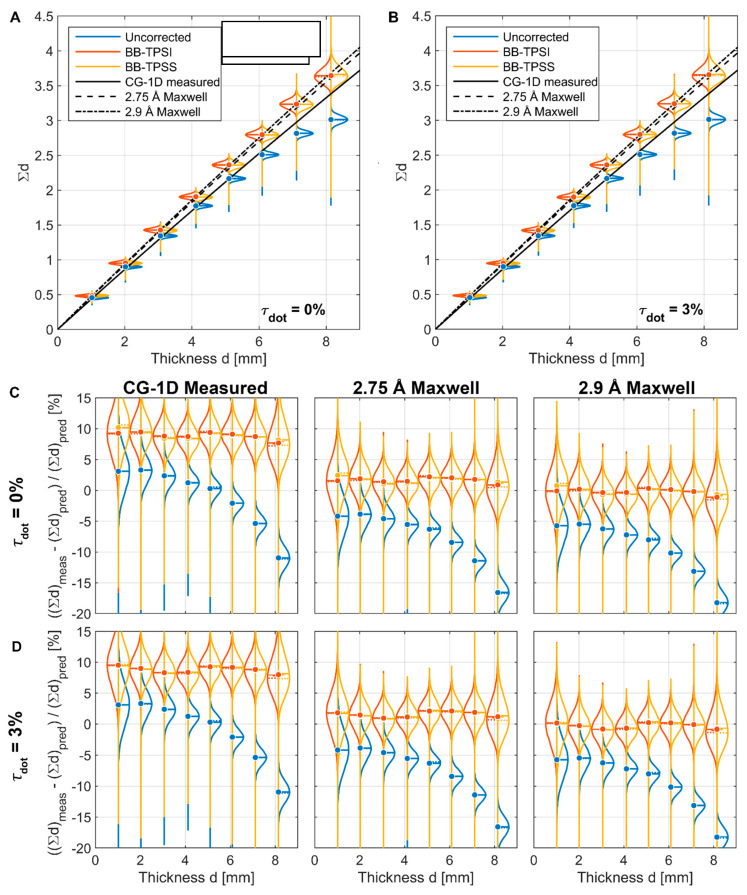
Comparison of BB scattering corrections for PMMA step wedge measured at CG-1D against predicted values of Σd using known cross sections [17] and the CG-1D spectrum as measured in 2010 and 2.75 Å and 2.9 Å Maxwell–Boltzmann spectra. (**A**) Values of Σd assuming 0% BB transparency. (**B**) Values of Σd assuming 3% BB transparency. (**C**). Relative Σd error assuming 0% BB transparency. (**D**). Relative Σd error assuming 3% BB transparency.

**Figure 15 jimaging-08-00164-f015:**
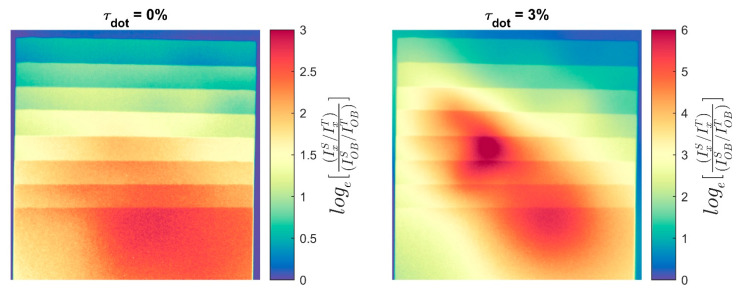
Log-normalized scattering-to-transmission ratios for PMMA step wedge at CG-1D.

**Figure 16 jimaging-08-00164-f016:**
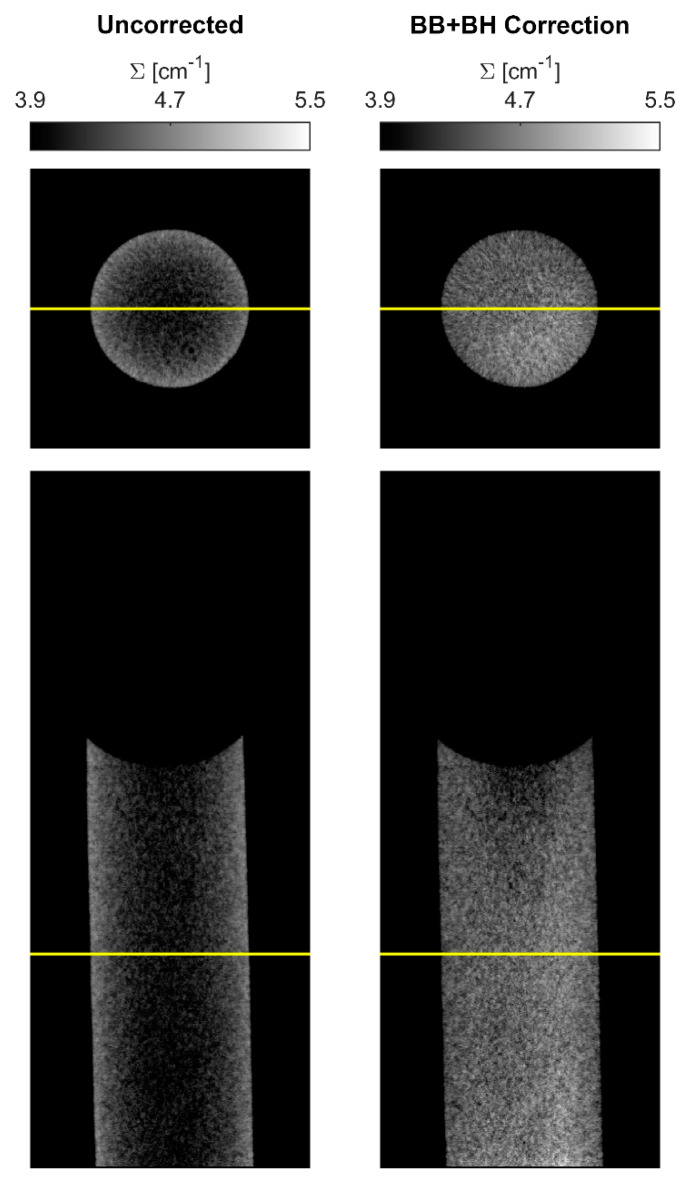
Slices of the water column CT reconstructions. Uncorrected CT shows significant cupping artifacts due to scattering and beam hardening. Combined BB and BH corrections (assuming $\tau_{dot}=0\%$ and 2.75 Å Maxwell–Boltzmann spectrum) effectively flatten the attenuation coefficient.

**Figure 17 jimaging-08-00164-f017:**
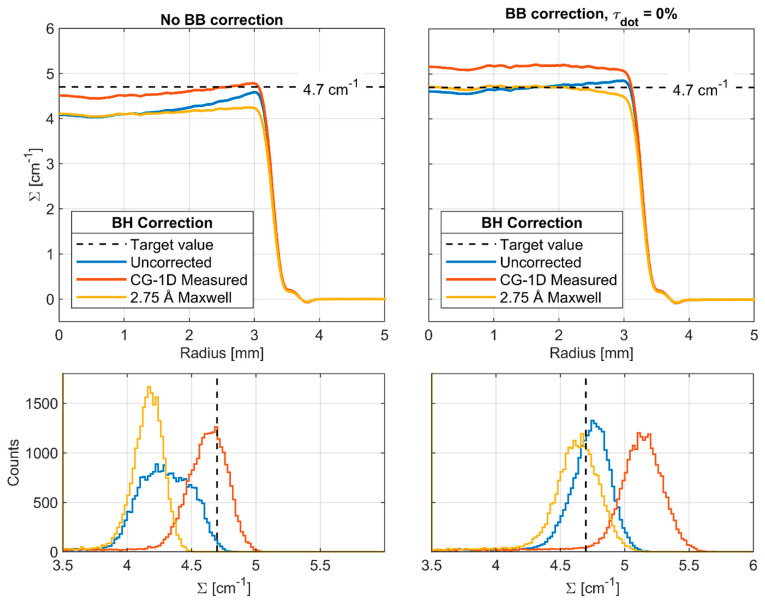
Water column CT radial profiles and histograms with BB and BH corrections.

**Figure 18 jimaging-08-00164-f018:**
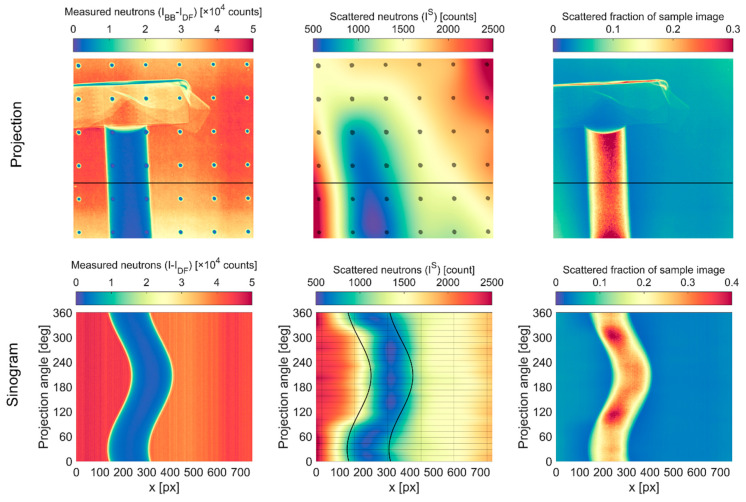
Scattering interpolation artifacts in CG-1D water column images. Images are pseudo-colored to enhance visual contrast. Horizontal line in projection images indicates slice used for sinogram images. Vertical lines in scattered neutron sinogram indicate BB locations, while horizontal lines indicate projection angles where BBs were used.

**Figure 19 jimaging-08-00164-f019:**
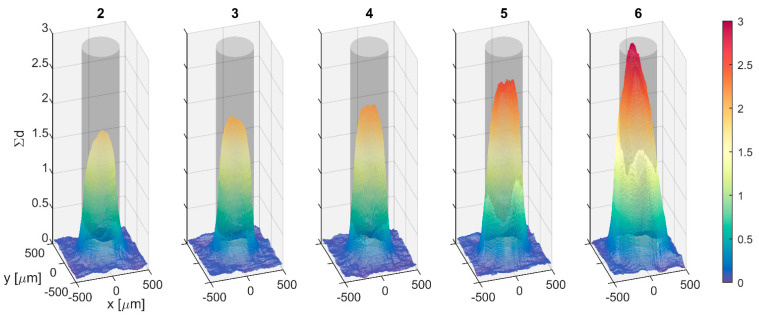
Representative dot from each 500 µm pattern measured at BT2 shows progressive mounding and trend toward conical shape with increased number of print layers. Transparent cylinder is target geometry.

**Figure 20 jimaging-08-00164-f020:**
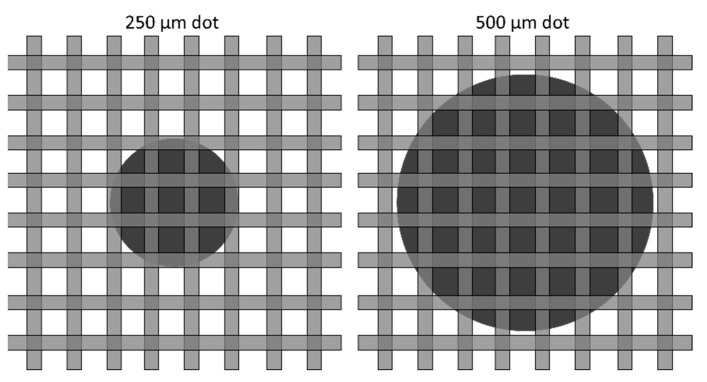
250 µm and 500 µm dots compared to 325 mesh. Dots and mesh are shown to scale.

**Table 1 jimaging-08-00164-t001:** Required Gd_2_O_3_ thickness (µm) to achieve given attenuation.

Spectrum	HFIR CG-1D (2.75 Å)		NCNR BT2 (1.8 Å)	
	99%	99.9%	99%	99.9%
Peak wavelength (mono)	28	42	39	58
Maxwell–Boltzmann (poly)	31	54	79	259
Measured (poly)	37	81	-	-

**Table 2 jimaging-08-00164-t002:** Print thickness for each pattern with increasing number of print layers (µm).

Number of Print Layers	250 µm × 2.5 mm	500 µm × 5 mm
2	19.2	43.6
3	25.5	52.0
4	30.3	63.0
5	30.5	79.6
6	32.0	80.0

**Table 3 jimaging-08-00164-t003:** Polynomial coefficients for beam hardening correction, $p\left(x\right)=\overset{n}{\underset{i=0}{\sum }}a_{i}x^{i}$.

Spectrum	$a_{0}$	$a_{1}$	$a_{2}$	$a_{3}$
CG-1D: measured	0	0.9942	0.0277	0.0006
CG-1D: 2.75 Å Maxwell	0	0.8501	0.0449	−0.0009

## Data Availability

Data collected at HFIR CG-1D and NCNR BT2 are available at https://doi.org/10.13139/ORNLNCCS/1870689 (accessed on 20 March 2022).

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
