# Peer review of "Fabrication of Black Body Grids by Thick Film Printing for Quantitative Neutron Imaging"

_2313-433X, 2022, doi:10.3390/jimaging8060164_

Round 1

Reviewer 1 Report

In this manuscript, the authors study a method for fabricating black body grids by thinck film printing of Gd2O3, which allows to measure neutron imaging. They analyze its performance with variation in feature size and number of layers with cold and thermal neutrons.   I have carefully read the manuscript and I think it is a valuable analysis of a promising method to quantify the mentioned neutron imaging. I find only a particular question that I would like the authors to comment in more detailed. I think it would be very interesting if the authors can be more precise about how to reproduce more accurately the desired geometry. I understand that the optimization of the method is beyond the scope of the work, but I think a deeper discussion would be increase the interest of the manuscript for the reader.

Author Response

The authors wish to thank the reviewer for their time and care in reviewing the manuscript. Regarding the question of how to reproduce the geometry more accurately, we list several possibilities in the Discussion, including optimization of the printing process (screen size, emulsion thickness, paste properties, print speed, and snap-off distance) and the possibility of incorporating post processing (lapping and laser trimming) at the end of the print or even between print layers. However, we have not yet pursued such an optimization, so any further discussion on the best approach or the smallest achievable feature size would be speculation.

Reviewer 2 Report

The paper presents a technique to rapidly and flexibly print black bodies grids with different parameters (pitch, size and thickness) to potentially tailor the grid to each specific experiment, as well as the characterization of printed grids themselves, both in radiography and tomography.
The manuscript is very well written and very exhaustive, so much so that I would have suggested to split it into two papers, one regarding the printing technique itself and one with the corresponding neutronics characterization. In any case, the authors' decision not to do so made this paper exceptionally thorough and detailed. It is very well referenced and clear so I have no problem recommending its publication.
The only very minor stylistical point I have, would be to modify the figures like fig. 13 and on to show points with suitable error bars rather than the complete distribution. This, I believe, would improve readability. I would leave this choice to the authors though, please consider this just a suggestion.

Other than that, ocngratulations to the authors for the excellent work.

Author Response

We thank the reviewer for their thorough review and kind remarks. Regarding the presentation of Fig 13 and others, we prepared many variations of these to find the best balance of accurately conveying the information while not visually overloading the figures. One of the shortcomings of using simple error bars is that they do not convey the asymmetry or overall shape of the underlying distributions, which is something we felt important to capture. By using half violin plots (kernel density estimates) with indicators of the mean, median, and mode, we were able to convey this information while keeping the different data sets visually distinct. We agree that this is ultimately a stylistic choice and others may have a different preference for visualizing the data.